# Remote Sensing Temporal Reconstruction of the Flooded Area in “Tablas de Daimiel” Inland Wetland 2000–2021

**DOI:** 10.3390/s23084096

**Published:** 2023-04-19

**Authors:** Jesús Pena-Regueiro, Javier Estornell, Jesús Aguilar-Maldonado, Maria-Teresa Sebastiá-Frasquet

**Affiliations:** 1Research Institute for Integrated Management of Coastal Areas, Universitat Politècnica de València, C/Paraninfo, 1, 46730 Grau de Gandia, Spain; jepere@doctor.upv.es; 2Geo-Environmental Cartography and Remote Sensing Group, Universitat Politècnica de València, Camí de Vera s/n, 46022 Valencia, Spain; jaescre@cgf.upv.es; 3Institute for Water and Environmental Engineering, Universitat Politècnica de València, Camí de Vera s/n, 46022 Valencia, Spain; jeagmal@upv.es

**Keywords:** Sentinel-2, Landsat series, Tablas de Daimiel National Park, inland wetland, water remote sensing index

## Abstract

Tablas de Daimiel National Park (TDNP) is a unique inland wetland located in the Mancha plain (Spain). It is recognized at the international level, and it is protected by different figures, such as Biosphere Reserve. However, this ecosystem is endangered due to aquifer overexploitation, and it is at risk of losing its protection figures. The objective of our study is to analyze the evolution of the flooded area between the year 2000 and 2021 by Landsat (5, 7 and 8) and Sentinel-2 images, and to assess the TDNP state through an anomaly analysis of the total water body surface. Several water indices were tested, but the NDWI index for Sentinel-2 (threshold −0.20), the MNDWI for Landsat-5 (threshold −0.15), and the MNDWI for Landsat-8 (threshold −0.25) showed the highest accuracy to calculate the flooded surface inside the protected area’s limits. During the period 2015–2021, we compared the performance of Landsat-8 and Sentinel-2 and an R^2^ value of 0.87 was obtained for this analysis, indicating a high correspondence between both sensors. Our results indicate a high variability of the flooded areas during the analyzed period with significant peaks, the most notorious in the second quarter of 2010. Minimum flooded areas were observed with negative precipitation index anomalies since fourth quarter of 2004 to fourth quarter of 2009. This period corresponds to a severe drought that affected this region and caused important deterioration. No significant correlation was observed between water surface anomalies and precipitation anomalies, and the significant correlation with flow and piezometric anomalies was moderate. This can be explained because of the complexity of water uses in this wetland, which includes illegal wells and the geological heterogeneity.

## 1. Introduction

It is a fact that wetlands play a key role in the hydrological cycle, and the hydrological conditions have a vital role on the ecological status of wetlands [1]. However, human activity causes changes in this cycle, and is one of the main causes of wetland degradation. Activities such as land reclamation for agriculture, dams, drainage, or surface water diversions cause the lowering of groundwater tables, the disappearance of springs, and the decrease in flooded areas in wetlands, that can even end in the full desiccation triggering spontaneous combustion of peatlands [2,3,4,5].

The role of flooding extent and duration is crucial for biodiversity (wildlife habitat) and other wetland ecosystem services (e.g., carbon storage, water quality, storing, floodwater, and maintaining the water levels in the dry season) [6]. In ecology, historical trends and observation of long-term change is central to understanding [7]. So, to select the most adequate conservation and restoration measures of wetlands, and to monitor their effectiveness, we require information about their hydrological conditions. Typically, hydrological monitoring of wetlands is conducted by in situ measures (piezometers and gauging stations) that may provide good temporal resolution but over a limited number of observation points [1,8]. However, gauge measurements offer little information to detect spatial patterns, such as flooding status, because usually distance between gauges is several kilometers or even more [1,8,9,10]. So, only with in situ measures, we cannot analyze the surface changes in these highly dynamic ecosystems.

A common approach to assess the spatial and temporal variability of water bodies is to use long-term (>25–30 years) remote sensing datasets. Satellite imagery notoriously improved water bodies monitoring in terms of detecting changes over time and space [11]. The advances in remote sensing technology allowed diverse hydrological applications, such as monitoring fluctuations of lake or dam surface or mapping small water bodies [12,13,14,15,16]. Remote sensing data from Landsat satellites (Landsat 5 (TM), Landsat 7 (ETM+), and Landsat 8 (OLI)) were widely used since the launch of the Landsat thematic mapper (TM) on 1 March 1984). However, Landsat has significant drawbacks to monitor wetlands. The Landsat spatial resolution of 30 m is unsuitable to monitor water bodies with surface area 0.1–5 ha (surface area monitoring error >20% or higher) [15,17]. Water surface classification accuracy varies according to the water body size and shape complexity [15]. This is especially relevant in arid and semiarid regions [11]. These regions are frequently affected by droughts, and this causes a more intensive use of groundwater that worsens the problems caused by the absence of precipitation. This turns into smaller unconnected water bodies than in other climates, and spatial resolution is a key parameter [15]. Additionally, the 16-day temporal resolution of these sensors can be a limiting factor to interpret wetland dynamics [13,15].

The Sentinel-2 satellite launched by the European Space Agency (ESA) in 2015 includes a constellation of two polar-orbiting satellites with 5-day temporal resolution and a maximum of 10 m spatial resolution [13,15,18]. The higher resolution of Sentinel-2, both spatial and temporal, can significantly improve the monitoring range, especially for small sites that cannot be covered by the Landsat satellite [8,11,13,15,18]. Some studies assessed a multi-sensor approach that takes advantage of Landsat and Sentinel-2 [12,15,16]. The combined Sentinel-2 and Landsat dataset theoretically allows for reduction in the temporal resolution to about three days [16]. The use of both satellites allows for reconstruction of a longer study period to detect trends and analyze the effect of restoration measures or climatic variability. To correctly interpret long data, it is necessary first to compare both sensors for the same period and area.

From these remote sensing images, we can extract water pixels through different methodologies. Among these, water indices are one of the most widely used because they are more reproducible and then more generalizable [13]. Commonly applied indices include the normalized difference water index (NDWI), modified normalized difference water index (MNDWI), normalized difference moisture index (NDMI), normalized difference vegetation index (NDVI), and automated water extraction index (AWEI) [8,14,18]. By selecting the appropriate threshold for these indices, the image pixels can be categorized into water or non-water, and wetland inundation conditions can be mapped.

The main objectives of this research are to reconstruct the flooded area in the Tablas de Daimiel National Park (TDNP) wetland for the 2000–2021 period using Landsat and Sentinel-2 sensors, and to assess the TDNP state through an anomaly analysis of the total water body surface. The anomaly of the total water body surface will be related to precipitation, river flow, and piezometric level anomalies to discern the variable that explains most of this semi-arid region wetland. The flooded area will be calculated by first choosing the optimal water remote sensing index and threshold for each sensor.

## 2. Materials and Methods

### 2.1. Study Area

Tablas de Daimiel is a floodplain wetland in central Spain, in the confluence of the Cigüela and Guadiana Rivers, in the municipalities of Daimiel and Villarrubia de los Ojos, within the province of Ciudad Real (439,400 m, 4,333,500 m, zone 30, UTM, European Terrestrial Reference System 1989, Figure 1). It is located at an altitude of 617 m. Climate in the region is typically continental and semiarid, characterized by a low average annual rainfall and recurrent droughts. Data from the meteorological station of Las Tablas de Daimiel show that the average annual rainfall is 376 mm for the historic register from 2001/02 to 2021/22, with a wide range of variation: a minimum of 182 mm, in the hydrologic year 2004/05, and a maximum of 589.8 mm, in 2009/10.

Tablas de Daimiel was declared a natural park (TDNP) by Decree 1874/1973 of 28 June; also, it is a biosphere reserve since 1981, and it is included in the Ramsar Convention on Wetlands since 1982. It was declared a special protection area (SPA) by Decree 82/2005 of 12 July and a special area of conservation (SAC) by Decree 187/2015 of 7 August. (SiteCode: ES0000013 of the Natura 2000 Network).

The surface of the TDNP is 3030 ha, of which approximately 1800 ha are subjected to flooding, with an average water depth of 0.90 m [7]. Under natural conditions, the wetland received inflows from the overflow from the rivers Guadiana and Cigüela and by contributions from groundwater [2,19]. Cigüela River provided brackish water on a seasonal basis. Since 1988, water from the Tagus River basin is diverted into the National Park through the Cigüela River [2,19]. Guadiana River and the Mancha Occidental aquifer discharged freshwater. The thickness of this groundwater system exceeds 400 m in some sectors [4].

Groundwater extraction for irrigation in the upper Guadiana River basin affected this system since the early seventies of the last century [7,19]. By the end of that decade, extractions exceeded TDNP inflows for the first time [2,20]. Between 1974 and 1987 the irrigated area increased from 30,000 to 125,000 ha, so in 1987, groundwater extraction nearly doubled the aquifer’s replenishment rate on a consistent basis (580 mm^3^/year vs. 320 mm^3^/year) [21,22]. The intensive groundwater use together with drought periods caused groundwater levels to drop drastically, and the decrease in the flooded surface of TDNP. In 1986, the TDNP was completely dry and burned a third of the surface [21,22]. The degradation process of this ecosystem caused again the drying up of the wetland from 2004 to 2009, and a smoldering peat fire started inside the TDNP in August 2009 [21,23].

### 2.2. Image Processing

Sentinel-2A/B images processed at level 1C were obtained from Copernicus (https://scihub.copernicus.eu/dhus/#/home, accessed on 1 June 2022) and they were atmospherically corrected with Sen2Cor tool (version 02.05.05). Landsat images TM Collection 2 Level-2 were obtained from USGS servers: Landsat-8 OLI (https://espa.cr.usgs.gov/index/, accessed on 1 June 2022) and Landsat-7 ETM+ and Landsat-5 (https://earthexplorer.usgs.gov/ accessed on 1 June 2022).

Images of high spatial resolution were used for validation. These images were obtained from the Spanish National Cartography Institute (IGN) Orthophoto 2017 and 2018 CC BY 4.0 © (spatial resolution 0.25 m, https://www.ign.es/web/ign/portal, accessed on 11 April 2023) and Google Earth ©. The dates of these images were the closest to Sentinel-2A/B, Landsat-8 OLI, and Landsat-5 TM image acquisitions. The full list of images used in this study by date is provided in Appendix A, Table A1.

The official cartography of this protected area (Natural Parks Net, Ministry for the Ecological Transition and the Demographic Challenge) was used to delimitate the wetland (Figure 1). We delimited the water and non-water polygons for each validation image. These polygons were delineated through visual examination using high-resolution images (orthophotos) as a base map, and it was conducted with the software ArcGIS 10.5 (ESRI 2016. ArcGIS Desktop: Release 10.5 Redlands, CA: Environmental Systems Research Institute). The visual delimitation was possible considering the high spatial resolution of the orthophotos (0.25 m). Water polygons smaller than 100 m^2^ were excluded for Sentinel-2A/B analysis considering the maximum spatial resolution of Sentinel-2A/B bands used in this study. For Landsat-8 OLI and Landsat-5 TM analysis, water polygons smaller than 900 m^2^ were excluded considering the spatial resolution of these sensors.

Six water indices, based on spectral information, were calculated using Sentinel-2A/B, Landsat-8 OLI, Landsat-7 ETM+, and 5 TM images according to their availability to cover the studied period (2000–2021). To this end, the equations shown in Table 1 were applied to this wetland (Figure 1). The choice of indices was based on literature review [8]. Once all these indices are calculated, their pixel values are classified into water/non-water classes using a threshold value that requires it to be evaluated since different authors propose different values for the same indices. We aimed to define a unique threshold for each sensor with the optimum results for the analyzed period. So, for each date and sensor with available images, we tested the thresholds from −0.50 to 0.50 with a 0.05 step, except for the automated water extraction index, no shadow (AWEI(NSH)) and the automated water extraction index, shadow (AWEI(SH)) indices, whose thresholds ranged from −50 to −5000, and the step is detailed in the results section.

To validate the results obtained from the Sentinel-2A/B, Landsat-8 OLI, and Landsat-5 TM images, we designed a random sampling of 60 points (30 water/30 non-water) for each sensor and date of the high spatial resolution images used in this study (see Appendix A, Table A1). Landsat-5 results were validated using three orthophotos (03/01/2005, 24 and 28/07/2006, 13 and 14/07/2009–31/08/2009). For Landat-8 and Sentinel-2A/B, other three orthophotos (27 and 28/06/2015, 13/11/2015, 28/09/2018) were used. A total of 540 points were considered for validation (180 points for each sensor imagery).

The ground control points were distributed randomly in the entire wetland surface. We selected the number of points according to the general guideline provided by Congalton and Green [24], who recommended a minimum of 50 samples for each map class for maps of less than 1 million acres in size and fewer than 12 classes. For the points sampled for each sensor (180), we compared the classification of each index (six indices in Table 1) and each threshold, with the ground truth images, to assess correct classifications. Overall accuracy index was calculated for each random sampling. [24]. The best index and threshold were selected according to overall accuracy results [8,25].

**Table 1 sensors-23-04096-t001:** Calculated spectral indices.

Index	Equation	Source	Sentinel-2 Bands
**NDWI**	[(GREEN − NIR)/(GREEN + NIR)]	[26]	[(B03 − B08)/(B03 + B08)]
**MNDWI**	[(GREEN − SWIR2)/(GREEN + SWIR2)]	[27]	[(B03 − B11)/(B03 + B11)]
**CEDEX**	(NIR/RED) − (NIR/SWIR)	[28]	(B05/B04)–(B05/B11)
**RE-NDWI**	[(GREEN − MIR)/(GREEN + MIR)]	[29]	[(B03 − B05)/(B03 + B05)]
**AWEI(SH)**	BLUE + 2.5 × GREEN − 1.5 × (NIR + SWIR) − 0.25 × SWIR	[30]	[B02 + 2.5 × B03 − 1.5 × (B08 + B011) − 0.25 × B12]
**AWEI (NSH)**	4 × (GREEN-MIR) − (0.25 × NIR + 2.75 × SWIR)	[30]	[4 × (B03-B11) − (0.25 × B08 + 2.75 × B12)]
**Index**	**Equation**	**Source**	**Landsat-8 OLI**
**NDWI**	[(GREEN − NIR)/(GREEN + NIR)]	[26]	[(B03 − B05)/(B03 + B05)]
**MNDWI**	[(GREEN − SWIR1)/(GREEN + SWIR1)]	[27]	[(B03 − B06)/(B03 + B06)]
**CEDEX**	(NIR/RED) − (NIR/SWIR)	[28]	(B05/B04) − (B05/B06)
**RE-NDWI**	[(GREEN–RED)/(GREEN + RED)]	[29]	[(B03 − B04)/(B03 + B04)]
**AWEI(SH)**	BLUE + 2.5 × GREEN − 1.5 × (NIR + SWIR1) − 0.25 × SWIR2	[30]	[B02 + 2.5 × B03 − 1.5 × (B05 + B06) − 0.25 × B07]
**AWEI (NSH)**	4 × (GREEN − SWIR1) − (0.25 × NIR + 2.75 × SWIR2)	[30]	[4 × (B03 − B06) − (0.25 × B05 + 2.75 × B07)]
**Index**	**Equation**	**Source**	**Landsat-5 TM**
**NDWI**	[(GREEN − NIR)/(GREEN + NIR)]	[26]	[(B02 − B04)/(B02 + B04]
**MNDWI**	[(GREEN − SWIR1)/(GREEN + SWIR1)]	[27]	[(B02 − B05)/(B02 + B05]
**CEDEX**	(NIR/RED) − (NIR/SWIR)	[28]	[(B04/B03) − (B04/B05]
**RE-NDWI**	[(GREEN − RED)/(GREEN + RED)]	[29]	[(B02 − B03)/(B02 + B03]
**AWEI(SH)**	BLUE + 2.5 × GREEN − 1.5 × (NIR + SWIR1) − 0.25 × SWIR2	[30]	[B01 + 2.5 × B02 − 1.5 × (B04 + B05) − 0.25 × B07]
**AWEI (NSH)**	4 × (GREEN − SWIR1) − (0.25 × NIR + 2.75 × SWIR2)	[30]	[4 × (B02 − B05) − (0.25 × B04 + 2.75 × B07)]

We made a temporal reconstruction of water body surfaces in TDNP since January 2000 to October 2021, with the best index using Landsat-8 OLI, Landsat-7 ETM+, and Landsat-5 TM (Table A1). Sentinel-2 A/B was also used for the period June 2015 to October 2021. We compared the results of water surfaces derived from Landsat-8 and Sentinel-2 for the period June 2015 to October 2021. This analysis is important to evaluate the performance of both sensors and to see if we can effectively increase temporal resolution by completing time series with images from both sensors.

To better assess the TDNP state, we made an anomaly analysis of the total water body surface along the study period using the data obtained from Landsat-8 OLI, Landsat-7 ETM+, and Landsat-5 TM series.

Surface data were averaged by quarter calculating the water surface anomaly index IAW_i_ (Equation (1)).
(1)$${IAW}_{i}=\frac{A_{WAi-}A_{WAaverage}}{A_{WAsd}}$$

*IAW_i_* being the index of the water surface anomaly for a quarter *i*, *A_WAi_* is the average water surface in the quarter *i*, *A_WAaverage_* is the average water surface in the quarter of the period analyzed (January 2000 to October 2021), and *A_WAsd_* is the standard deviation of the water surface in the quarter of the period analyzed. Based on these values, the quarters with positive and negative anomalies were analyzed.

Precipitation data from the meteorological station of Las Tablas de Daimiel were obtained (https://eportal.mapa.gob.es/websiar/SeleccionParametrosMap.aspx?dst=1, accessed on 11 April 2023). With these values, the precipitation anomaly index was calculated, *IAP_i_* (Equation (2)) [8,31].
(2)$${IAP}_{i}=\frac{P_{i-}P_{average}}{P_{sd}}$$

*IAP_i_* being the index of the precipitation anomaly for a quarter *i*, *P_i_* is the average precipitation in the quarter i, *P_average_* is the quarter average precipitation in the period analyzed (hydrologic year 2001 to 2021), and *P_sd_* is the standard deviation of precipitation in the quarter of the period analyzed.

Flow data were obtained from the Guadiana River gauge in the Ruidera municipality, 15 km upstream of TDNP (https://sig.mapama.gob.es/redes-seguimiento/?herramienta=aforos, accessed on 11 April 2023) (Figure 1). The flow anomaly index was calculated, IAFi (Equation (3)).
(3)$${IAF}_{i}=\frac{F_{i-}F_{average}}{F_{sd}}$$

*IAF_i_* being the index of the flow anomaly for a quarter *i, F_i_* is the average flow in the quarter *i*, *F_average_* is the quarter average flow in the period analyzed (hydrologic year 2000 to 2021), and *F_sd_* is the standard deviation of flow in the quarter of the period analyzed.

Finally, the piezometric data were obtained from 4 piezometers surrounding TDNP (Figure 1). The data used for the analysis was water depth. Piezometers European codes were: ES040ESBT000404042, ES040ESBT000404046, ES040ESBT000404145, and ES040ESBT000404294, and were all located in the Mancha Occidental I aquifer (https://sig.mapama.gob.es/redes-seguimiento/?herramienta=Piezometros, accessed on 11 April 2023). With these values, the piezometric anomaly index was calculated, IAPZi (Equation (4)).
(4)$${IAPZ}_{i}=\frac{{PZ}_{i-}{PZ}_{average}}{{PZ}_{sd}}$$

*IAPZ_i_* being the index of the piezometric anomaly for a quarter *i, PZ_i_* is the average piezometric level in the quarter *i*, *PZ_average_* is the quarter average piezometric level in the period analyzed (hydrologic year 2009 to 2021), and *PZ_sd_* is the standard deviation of piezometric level in the quarter of the period analyzed.

Spearman correlation coefficient was calculated between the anomalies of the following variables, water surface, precipitation, flow, and piezometric level by quarters.

## 3. Results

Extraction of water bodies was obtained from Sentinel-2 A/B, Landsat-5, Landsat-7 ETM+, and Landsat-8 images (Table A1). To do this, six water indices (Table 1) and a set of thresholds were analyzed. For Landsat-5 images, the most accurate results were obtained when the MNDWI (0.88) and AWEI (nsh) (0.88) water indices were selected applying the thresholds values of −0.15 and −900, respectively (Figure 2a). For Landsat-8 images, the maximum overall accuracy was obtained for MNDWI (0.99) using a threshold value of −0.25 and −0.30 (Figure 2b). For these images, accurate results were also obtained for AWEI(nsh) (0.98), AWEI(sh) (0.97), and NDWI (0.96) water indices (Figure 2b). For AWEI (nsh) the highest values of overall accuracy were obtained for the thresholds −4000, −4500, and −5000 (Figure 2b). For the AWEI(sh) water remote sensing index, the selected threshold was −2000 and for NDWI −0.25 and −0.30 (Figure 2b). For Sentinel-2 images, high performances in terms of overall accuracy was found for NDWI (0.99, threshold −0.20), MNDWI (0.98, thresholds −0.05 and −0.1), AWEI (sh) (0.98, thresholds −800, −900, −1000, and −1500), and AWEI(nsh) (0.98, thresholds −1000 to −4500) (Figure 2c). From these results, the MNDWI index with a threshold of −0.15 was selected for Landsat-5 images to extract water bodies. Although the same global accuracy was obtained for AWEI(nsh) index and threshold −900, the former index was selected since a lower performance was observed in areas close to the border of the water polygons. For Lansat-8 images, the MNDWI index was selected and a threshold of −0.25. The threshold value of −0.30, which generated the same accuracy, was disregarded since the same lower performance was observed in areas close to the border of water surfaces. For Sentinel-2 images, the NDWI index and a threshold of −0.20 were selected for extracting water bodies. Water areas were computed for each analyzed image in the period 2000–2021 using these indices and thresholds. Then, the water surface anomaly index was calculated for the analysis period (Equation (1)).

The water surface anomaly index showed several patterns in the analyzed period (Figure 3a,b). From 2001 to 2003 positive and negative anomalies were detected without a clear trend. Between the second quarter of 2004 and the first quarter of 2005, positive anomalies were obtained. From this quarter to the first quarter of 2006, anomalies close to 0 were observed. Then, a longer period (3t-2006 to 1t-2010) with relevant negative anomalies is shown. After this phase, significant positive anomalies were observed until the third quarter of 2013, when a period with anomalies close to 0 was observed. This trend continued until the second quarter of 2019, when a set of significant negative anomalies was detected until 2021. In Figure 3, the precipitation anomaly index (Equation (2)) is also represented. Although some quarters showed positive and negative anomalies concurrent with water surface anomaly, other quarters showed opposite behavior for the anomaly indices (47% of the quarters) and absolute values of anomaly were significantly different in almost all quarters. In this sense, Spearman correlation analysis between these two parameters was calculated obtaining a value of −0.049 and *p*-value 0.677, indicating a lack of a significant statistical relationship between them.

In Figure 4a comparison among the water surface anomaly index (Equation (1)) and the flow anomaly index (Equation (3)) is shown. In this case, concurrent positive and negative anomalies of these two indices were observed for more than 50% of the analyzed quarters. In this context, negative values of IAWi and IAFi were observed for the period 3t-2006 to 1t-2009 and positive values from 2t-2010 to 1t-2012. In this case the Spearman correlation analysis generated a value of 0.369 and a significative *p*-value (Table 2), indicating, although in a moderate way, the existence in some periods of a statistical relationship between the water surface and river flow anomalies.

In Figure 5 a comparison among the water surface anomaly index (Equation (1)) and piezometric anomaly index (Equation (4)) is shown. In this case, equal distribution patterns were detected in some quarters for the analyzed period (2009–2020). The highest positive anomalies derived from the piezometric anomaly index indicate low availability of groundwater (highest groundwater depth). This occurred from the third quarter of 2009 to the first quarter of 2010. For this period, significant negative anomalies for water surfaces were observed. This same pattern was detected for the period of the second quarter to the fourth quarter of 2020. In contrast, the highest positive anomalies for water surfaces (third quarter of 2010–third quarter 2013) occurred during the lowest negative anomalies of piezometric index. The low piezometric anomaly index means that groundwater level is close to ground level. Similar patterns were observed from the second quarter of 2015 to the second quarter of 2016. For the rest of the quarters in the analyzed period, water surface anomalies close to 0 matched with negative anomalies for piezometric anomalies values, indicating that groundwater outcrops into the surface. The Spearman coefficient for these two anomalies was −0.406 and the obtained *p*-value indicates a moderate inverse statistical relationship between the piezometric level and water surface anomalies.

To compare the results of water surfaces derived from Landsat-8 and Sentinel-2, a linear regression model was calculated (Figure 6). An R^2^ value of 0.87 was obtained for this analysis, indicating a high correspondence between the water surfaces derived from both sensors.

## 4. Discussion

The selected indices were the MNDWI index with a threshold of −0.15 for Landsat-5 images, the MNDWI index and a threshold of −0.25 for Lansat-8 images, and the NDWI index and a threshold of −0.20 for Sentinel-2 images. These indices are among the most used for detecting water surfaces with both sensors, as they have a high ability to separate water zones from other coverages [6,14,18]. The NDWI index often identifies built-up areas as water. For solving this problem, the modified NDWI (MNDWI) is obtained using the MIR band instead of the NIR band [14,27]. Pena et al. (2020) [8] also found that the best performing index was the NDWI for Sentinel-2 images in Mediterranean wetlands characterized by the presence of small water bodies, although their results are optimal for the −0.30 threshold. The delimited natural area obtained from official cartography was used in both this research and Pena et al. (2020) [8], which allows for overcoming this NDWI limitation. Adequate thresholds are required for accurate water extraction, but there is no specific rule for setting the threshold, as it depends more on the region [14].

Climatic variability has a direct effect on surface water distribution, and the impacts of drought on surface water were well documented [6]. Droughts are becoming more common around the world [6], and to monitor their impact on the wetland flooded surface, it is necessary to compile information on water surfaces, and on meteorological, hydrological, and hydrogeological variables. In this research, we studied the evolution of water surface on a quarter basis thanks to remote sensing data, and we selected precipitation, river flow, and piezometric levels as representative variables of the hydrological cycle. The water surfaces estimated in this study provide evidence to understand the surface water changes of the TDNP wetland. Assessing the state of groundwater-dependent wetlands is complicated and more in arid and semiarid regions where recharge through precipitation may be concentrated in time [2]. The anomaly indices studied enable identification of the hydrological variables that have a major impact on water surface. Our results show a lack of a statistically significant relationship between water surface anomaly index and precipitation anomaly index. In contrast, a significant positive relation (0.369) between water surface anomaly index and flow anomaly index, and a significant negative relation (−0.406) between water surface anomaly index and piezometric anomaly index were found.

The absence of significant correlation between surface water cover and precipitation in wetlands was pointed out by other authors [18]. Tough as it may seem that there is no correlation between water surface and precipitation, we can observe some key moments in the historical evolution of TDNP in Figure 3. The most notorious is in the second quarter of 2010, when after the high positive precipitation anomaly registered in the first and second quarter of 2010 the water surface anomaly changed from negative to positive values. This positive water surface anomaly remained until the fourth quarter of 2016 when it showed an oscillating behaviour between negative and positives values for about a year and a half. This matches one of the periods when the TDNP was almost completely flooded from 2010 to 2013 [2,7], and the positive water surface anomalies were the highest. In this period, groundwater naturally began to discharge into the wetland and springs reappeared [2]. Prior to this wet period, we can observe in Figure 3 negative precipitation index anomalies since the fourth quarter of 2004 to the fourth quarter of 2009. This period corresponds to a severe drought that affected this region and caused important deterioration to the TDNP that ended in spontaneous peatland fires [2,23]. These fires were extinguished only after the 2010 flooding [2,23]. In Figure 7, we represent the evolution of water surface in TDNP during the driest period (September 2008 to November 2009), where we can observe the disappearance of surface waters in November 2009. Additionally in Figure 7, we show the recovery in 2010 and how water surface extension remained during 2011. Since 2014, another drought period started with almost all quarters showing negative precipitation index anomalies. However, water surface index anomalies started the negative trend in the second quarter of 2018. This lag between negative precipitation and water surface anomalies could be attributed to the resilience of the groundwater system, and the temporal sequence of droughts where the first drought that manifests itself is meteorological, then in surface water, and then in groundwater.

The correlation of the water surface anomaly index with flow anomaly index and piezometric anomaly index is similar but of an opposite sign. Flow data were obtained from the Guadiana River gauging station, which is the main contributor to TDNP and the historical data series is more complete than the Cigüela River. No significant correlation was observed between the precipitation anomaly index and the flow anomaly index (Table 2). Guadiana River flow highly relies also on groundwater discharges from the Mancha Occidental aquifer. Several factors can influence this absence of significant correlations. First, TDNP oscillations in water depth were attributed to preferential infiltration in some sectors due to the heterogeneity of geological formation in this region, which includes rapid circulation in karstified sectors and slow flow in areas with low-permeability sediments [2,3,32]. Additionally, the intensive groundwater pumping for irrigation was reported as the cause of a notorious dissociation between the surface and groundwater networks for much of the 1970–2014 period [2,4]. It is during unusually wet periods that significant reduction in groundwater extractions and increased groundwater recharge are observed [2]. In these periods, such as the period of 2010 to 2014 (Figure 5), water depth is very shallow and the springs outcrop maintaining a positive water surface anomaly.

With regards to spatial resolution of both sensors, we found an acceptable agreement between the estimated water surface between Landsat-8 and Sentinel-2 (R^2^ = 0.87) (Figure 6). In Figure 8, we represent the evolution of water surface during 2021 estimated with both sensors. We can observe the high similarity between both sensors’ estimation and the reduced water surface of this period showing the effects of the last dry years. The differences could be attributed to several factors. One important factor is changes in water surface due to different image acquisition data. These ecosystems are highly variable and water surface can change in a short time period, but other factors could be associated to the different spatial resolution. After a dry period, when groundwater starts to outcrop, unconnected puddles are formed that will connect to form bigger water bodies if favorable conditions remain [7]. Then, the higher spatial resolution of Sentinel-2 (10 m) can detect smaller water bodies than the Landsat-8 (30 m) [18] in the period analyzed.

## 5. Conclusions

The results of this study indicate the applicability of Landsat and Sentinel-2 images to evaluate the temporal variation of water bodies in the Tablas de Daimiel wetland. This study enabled detecting periods above-average water surfaces, and dry periods with flooded surfaces under average conditions. To this end, a preliminary study of water indices was applied to extract water bodies reporting different indices and thresholds for each sensor satellite. Precipitation, flow, and piezometric anomaly indices were compared to the water surface anomaly index, revealing a poor relationship between precipitation and water surface anomalies. In contrast, remarkable results were found for flow and piezometric anomalies observing significant relationships among them. The extraction of water surfaces using Sentinel-2 and Landsat-8 images was also compared. The results indicate a good correlation between these two sensors. Although the higher resolutions of Sentinel-2 images recommend their suitability compared to Landsat images, these results reveal the potential of Landsat images for analyzing historical periods of water surface. The information withdraw of this research contributes to monitoring the state of endangered wetlands, helping to adapt management plans that leads to a well-preserved state of conservation.

## Figures and Tables

**Figure 1 sensors-23-04096-f001:**
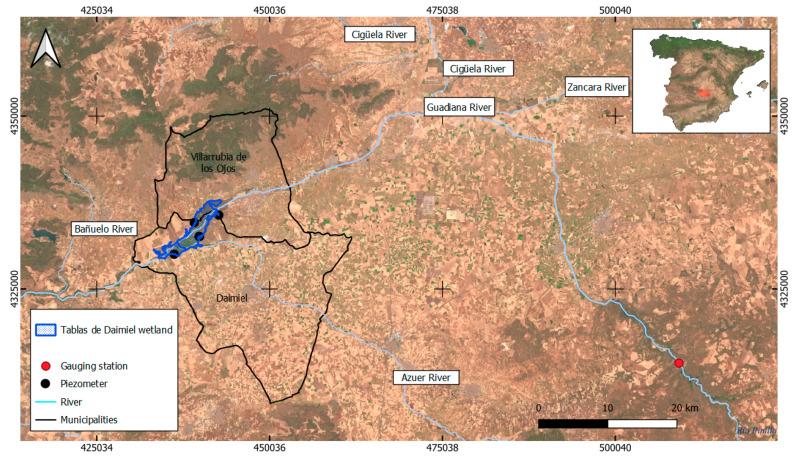
Study area including the perimeter of the Tablas de Daimiel National Park (blue polygon), gauging stations (red points), piezometers (black points), rivers (cyan lines), and municipalities in which the studied wetland is located (black polygons).

**Figure 2 sensors-23-04096-f002:**
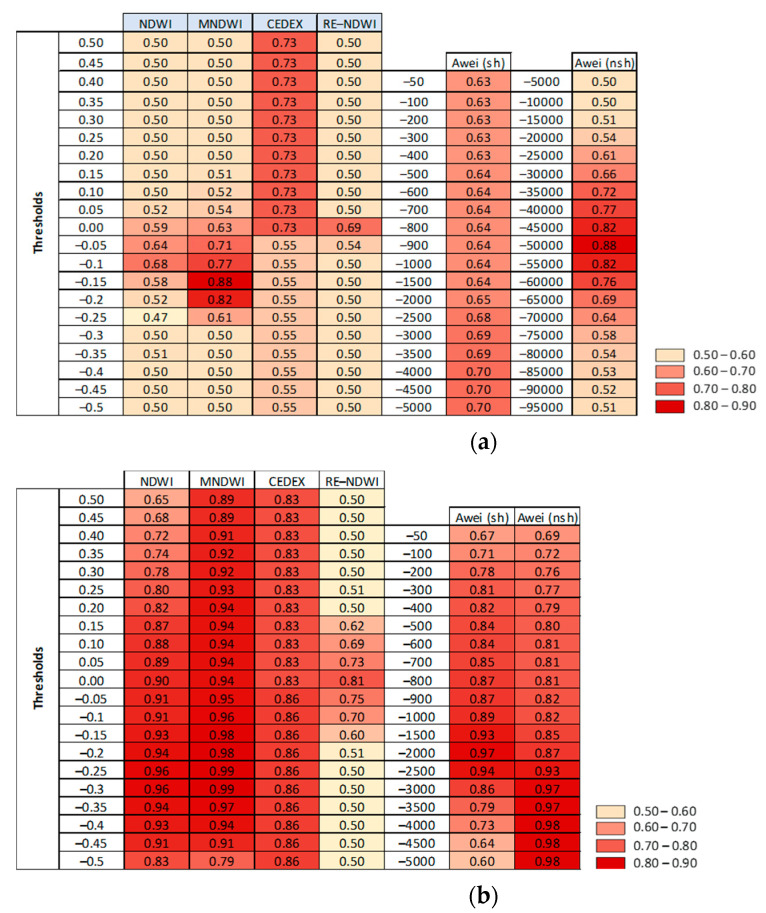
Overall accuracies for a set of thresholds applied to the Landsat-5 (**a**), Landsat-8 (**b**), and Sentinel-2 images (**c**). Light shaded colors show the indices and thresholds with poorest performance. Shaded in red colors are used for indices and thresholds with the most accurate (closer to 1).

**Figure 3 sensors-23-04096-f003:**
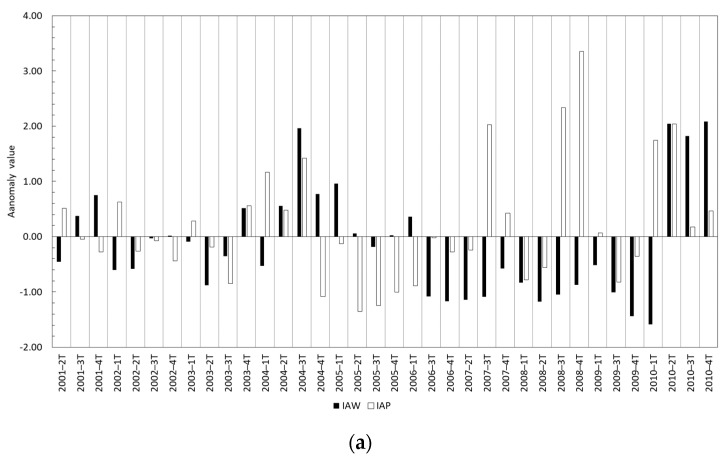
Water surface anomaly index (IAW) and precipitation anomaly index (IAP) in the period 2001–2010 (**a**) and 2011–2021 (**b**).

**Figure 4 sensors-23-04096-f004:**
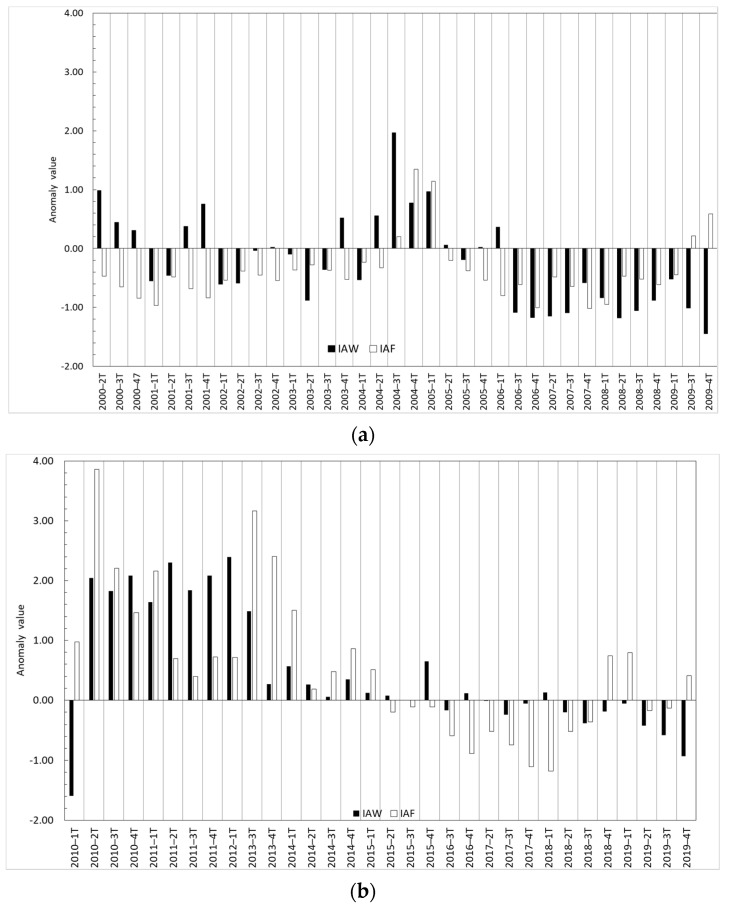
Water surface anomaly index (IAW) and flow anomaly index (IAF) in the period 2001–2010 (**a**) and 2011–2021 (**b**).

**Figure 5 sensors-23-04096-f005:**
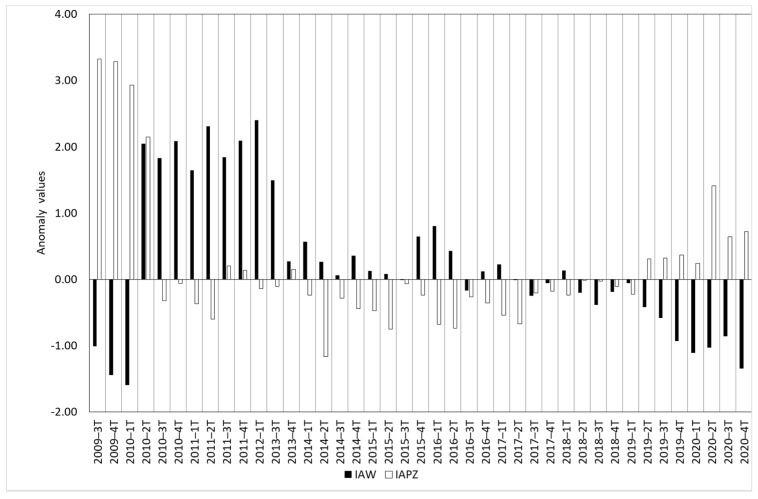
Water surface anomaly index (IAW) and the piezometric anomaly index (IAPZ) in the period 2009–2021.

**Figure 6 sensors-23-04096-f006:**
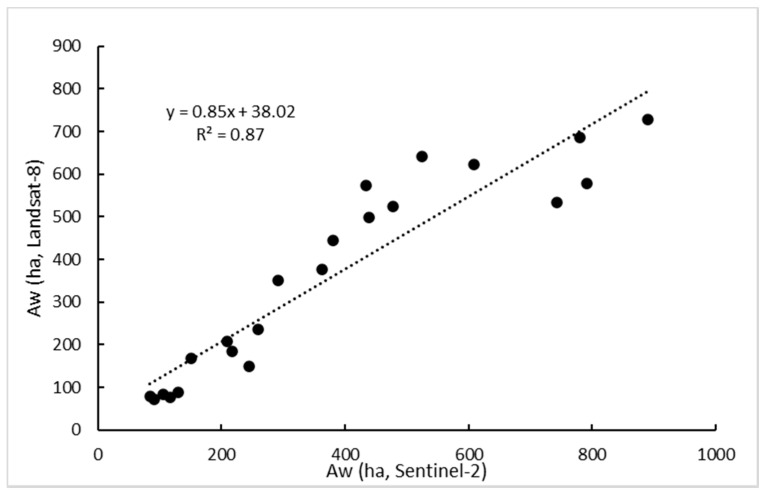
Comparison between water areas derived from Landsat-8 and Sentinel-2A/B images in the period 2015–2021.

**Figure 7 sensors-23-04096-f007:**
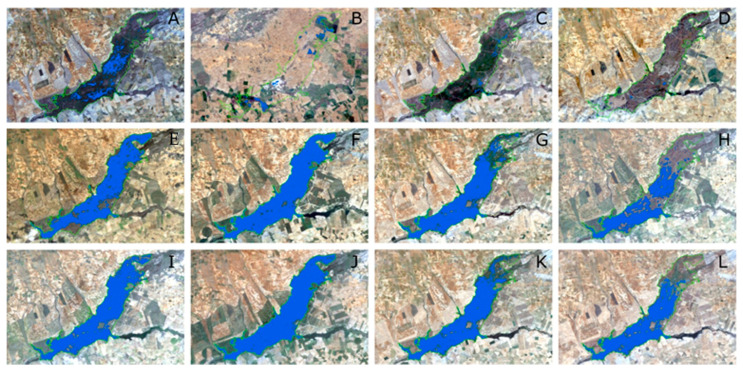
Water bodies in the Tablas de Daimiel National Park delimited according to the MNDWI spectral index. Landsat-5 TM images (threshold −0.15): (**A**) September 2008, (**B**) April 2009, (**C**) July 2009, (**D**) November 2009, (**E**) January 2010, (**F**) May 2010, (**G**) July 2010, (**H**) December 2010, (**I**) February 2011, (**J**) April 2011, (**K**) July 2011, and (**L**) October 2011.

**Figure 8 sensors-23-04096-f008:**
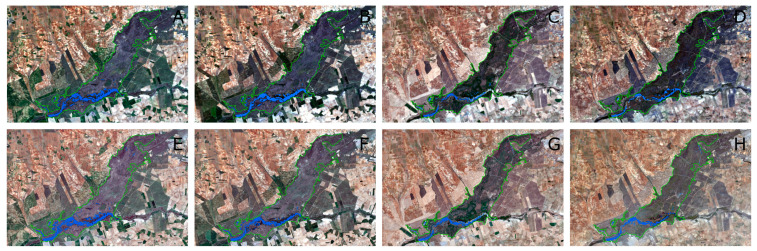
Water bodies in the Tablas de Daimiel National Park delimited according to the MNDWI spectral index for Landsat-8 OLI (threshold −0.25) and NDWI for Sentinel-2 A/B (threshold −0.20). Landsat-8 OLI images; (**A**) March 2021, (**B**) April 2021, (**C**) August 2021, and (**D**) October 2021. Sentinel-2 A/B images: (**E**) March 2021, (**F**) April 2021, (**G**) August 2021, and (**H**) October 2021.

**Table 2 sensors-23-04096-t002:** Spearman coefficient values among anomaly indices.

	IAW		IAF		IAF	
	Spearman Coefficient	*p*-Value	Spearman Coefficient	*p*-Value	Spearman Coefficient	*p*-Value
**IAP**	−0.049	0.677				
**IAF**	0.369	0.003				
**IAPZ**	−0.406	0.009			0.109	0.544
**IAP**			0.087	0.488		

## Data Availability

Not applicable.

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
