# Peer review of "Remote Sensing Temporal Reconstruction of the Flooded Area in “Tablas de Daimiel” Inland Wetland 2000–2021"

_sensors, 2023, doi:10.3390/s23084096_

Round 1

Reviewer 1 Report

I have reviewed the manuscript title "Remote sensing temporal reconstruction of the flooded area in “Tablas de Daimiel” inland wetland 2000-2021" which is an interesting topic for the readers community however I have following observations on the methodology opt and results generated. 

The abstract should be revised. Do not discuss the whole methodology in it. Try to discuss the results generated as well. 

Introduction part is well designed. 

Methodology section need to be revised and should not be emphasized on the technical details of software or products rather It should be focused on  methods opt.

The results section should be revised and the mapping product should be added as the satellite imagery was involved. It is suggested if the maps quantity is high than atleast the final 2021 and start 2000 map comparison should be presented.  

I don't feel to qualified to assess the quality of English in this paper but the English language and style are fine/minor spell check required.

The conclusion should be added. 

Author Response

Dear reviewer,

we attach a file with the full comments and answers to all the reviewers.

thanks for your help.

Reviewer 2 Report

Article Title: Remote sensing temporal reconstruction of the flooded area in “Tablas de Daimiel” inland wetland 2000-2021

It seems to me that the importance of the NDWI and MNDWI indices was not clearly demonstrated in the results. Have they not been correlated to precipitation, flow and piezometric data like the AWE? Indices from Table 1 such as CEDEX were never used?

The results focused only on the relationships between the water anomaly index (IAW) and the precipitation, flow and piezometric indices. If something is added in this regard, the discussion should be updated for these results.

Minor revisions:

- Line 164 to 173: explain in a more didactic way the limits used to classify water bodies;

- In the description of the terms of equation 1, on line 204, I believe that the numerator term AWAaverage was written as ALAaverage;

- Line 235: where is Table 2?

- About the result found from the relationship between IAW and IAP: the IAP of the quarter was correlated exactly with the IAW of the quarter, it would not be the case to investigate IAP with some lag, considering that the recharge of a swampy area can happen more slowly over the course of a longer rainy season? I ask this because the relationship between IAF and IAW was positive and significant (p-value = 0.003), and generally flows respond to rainfall with a lag of a few months.

- Would it be possible to enrich the work with images of the indices in important periods that were highlighted in the results?

Author Response

(The authors gave the same response as above.)

Author Response

(The authors gave the same response as above.)

Round 2

Reviewer 1 Report

Thank you very much for improving the quality of the manuscript. 

Author Response

Thanks for your appreciation. 

Reviewer 2 Report

The authors clarified my doubts and accepted the suggestions. I am satisfied.

Author Response

Thanks for your appreciation. 

Reviewer 3 Report

All questions are well addressed. Thanks

Author Response

Thanks for your appreciation.